# Molecular Signal Transfer of Highly Diluted Antibodies to Interferon-Gamma Regarding Kind, Time, and Distance of Exposition

**DOI:** 10.3390/ijms25010656

**Published:** 2024-01-04

**Authors:** Igor Jerman, Linda Ogrizek, Vesna Periček Krapež, Luka Jan

**Affiliations:** BION Institute, Stegne 21, 1000 Ljubljana, Slovenia; linda.ogrizek@bion.si (L.O.); vesna.pericek@bion.si (V.P.K.); luka.jan@bion.si (L.J.)

**Keywords:** ultra-high dilution, UHD signal transfer, physicochemical measurements, UV/VIS spectroscopy, distance

## Abstract

Physicochemical examinations of very high dilution (UHD) solutions subjected to certain physical factors (such as shaking) are becoming more frequent and are increasingly producing conclusive results. A much less studied phenomenon is the transfer of molecular information (i.e., UHD signals of dilute substances) from one liquid to another without an intermediate liquid phase. The aim of this study was to investigate the possibility of such a transfer of the UHD signal from the UHD solutions to the receiver solution, in particular, if the molecular source used in the donor solutions was the biologically active antibodies to interferon-gamma molecule. We were especially interested in how the transfer of the UHD signal is affected by the time of exposure of the receiver to the donor, the distance between the two, and how the transfer is affected by activation (striking) versus exposure alone. Signal transfer was evaluated by differential measurements of electrical conductivity, ORP, pH, and UV/VIS spectroscopy of the exposed liquid. The results showed that activation strongly influences signal transfer and that this can be compensated to some extent by prolonged direct exposure. In principle, exposure time has a positive effect on signal transfer. Interestingly, the results of different distances between the donor and receiver showed similar changes in the parameters in the range of 0–4 cm, as estimated in this study. While the study mainly confirms the two hypotheses, it also raises a number of new questions and provides clues for further research.

## 1. Introduction

Recently, the behavior of ultra-high dilutions (hence UHD), where the concentration of the original substance is many magnitudes below any possible chemical or biological expected effects, became a focus of numerous studies. Such aqueous solutions (liquids) could be considered as possessing molecular imprints of one or more compounds dissolved in the mother solution. These imprints will be named UHD signals or molecular information, and their further processing of UHD signal transfer. An extended pilot research into this topic was recently performed by the BION research team.

One part of this study, already published [1], addressed the potential detection of molecular signals from an initially diluted substance purportedly present in ultra-high dilutions within distilled water. This investigation employed three physicochemical methods (electrical conductivity, ORP, and pH) and UV/VIS spectroscopy. In this part of the study, we identified the signals (named UHD signals) in variously composed UHD solutions of antibodies to IFNγ (marked as a-IFNγ) or of distilled water, their further centesimal dilution and physical transfer of the signal mechanical means via glass.

In the subsequent phase of the research, our focus was directed toward specific time and distance parameters associated with the previously described physical transfer of UHD signals. This transfer process typically entails the activation (physical excitation) of the donor-side liquid (UHD solution). The activation is commonly achieved through either shaking or rhythmic striking on the glass in the receiver’s direction. A fundamental assumption, based on similar successfully conducted experiments by other groups and ours, suggests that with an appropriate activation of the donor UHD solution, the UHD signal can spread to nearby receiver liquids or solution(s) via glass or even the air in between. Therefore, in the basic experimental situation, we have three main elements of this transmission: (1) the activation of the donor, (2) the transfer itself through non-fluid media (glass, air, electromagnetic field…), and (3) the reception (imprinting) of the UHD signal in the receiver liquid (solution) [2].

The research methods that demonstrate the UHD signal via differences between processed solutions to equally treated water with no diluted substances (i.e., control) are mostly physicochemical, physical [3,4,5,6], and even biological [7,8,9,10]. The effects observed at extremely high dilutions cannot be attributed to chemistry but rather to the UHD signal, which appears to be imprinted and retained in the processed liquid or solution in a manner that is not yet fully comprehended. This holds particularly true for liquids or solutions into which the signal has been physically transmitted using the general procedure outlined earlier.

The exact nature of this signal remains unknown; however, given the theoretical considerations on the possible stability of UHD signals in water, the most plausible hypothesis is that the signal is received and stored by specifically ordered water molecular assemblies (coherent domains). Since, in general, they have a diameter of one to a few hundred nanometers, they compose the so-called mesoscopic water phase [11,12,13,14,15]. They were even observed via different methods, like in experiments by Konovalov and Ryzhkina [16] or Sedlák and Rak [17].

The most convincing theoretical model that covers the stable formations of mesoscopic water is based on quantum electrodynamics (QED) [18,19,20], even if there are also alternative explanations like Meesen’s nano-pearls [21] or clathrates [22]. For many experimental situations, QED cannot yet yield exact mathematical predictions of experimental results in processing the UHD signal, yet it has models that may well serve as a general theoretical explanation for understanding molecular imprinting (i.e., creation of the UHD signal) and the storage of the signal in a fluid medium via the already mentioned coherent domains of various kinds. The QED theory sees imprinting and storing of molecular information as a process analogous to that in materials consisting of domains with magnetic moments [23,24]. In theory, coherent domains possess ferroelectric properties due to dynamically ordered cold vortices and can thus constitute long-term information storage agents.

In our preceding article [1], we extensively elaborated on the rationale and context for our application of four specific research methods, including electrical conductivity (χ), pH, oxidation–reduction potential (ORP), and UV/VIS spectrometry. We wish to emphasize that these methods are designed to identify general physicochemical alterations arising from the presence of dynamically ordered water nanostructures linked to the UHD signal. These changes can result from high dilution combined with vigorous shaking or physical transfer, as described in this report. These structures have already been identified by various methods and labeled differently by different authors, like nano-associates [16], water clusters [25,26,27,28], coherent domains [29,30,31], nanobubbles [17,32], nanoparticles [33], or naneons [34]. While we delved into possible detection mechanisms in our previous publication [1], the current paper will primarily focus on the applied research methods in the Section 4.

Our overarching concept, as previously outlined in [1] and further elaborated upon in this second part, is grounded in the fundamental assumption that the UHD signal and its transmission extend beyond the dynamically organized water structures previously mentioned. Research conducted by independent teams globally has revealed that the UHD signal can also be transferred from one liquid (the donor) to another (the recipient) without direct physical contact between them [2,35,36,37]. From this viewpoint, the nature of the UHD signal seems complex and only to some measure associated with water material structures like coherent water domains, nano-associates, nanobubbles, clusters, etc. It seems that at least one aspect of the signal can transcend any water structure and can be transmitted through media such as glass, air, or magnetic fields. Yet, adhering to Ockham’s razor, we hypothesize that the UHD signal is a singular entity, whether residing in ordered water domains, in the air, or glass. It should represent a specific, presumably coherently oscillating energy or field structure coupled to mesoscopic structures within liquids for its habitat. Once certain physical conditions—such as the previously mentioned activation—are met, it may depart from its temporary environment within a liquid (or at least radiate from it) and may be transferred, absorbed, and stored by another liquid that is physically and chemically separated. As to the nature of the physical background of the UHD signal, it is generally presumed to be electromagnetic [37,38,39,40]. However, while it may manifest as coherent EM oscillations in mesoscopic water domains, outside of water, potentially, it could represent a quantum phenomenon or involve an as yet unknown quantum field or the field of quasi-particles. For instance, Kernbach proposes spin transfer [41,42], and some other authors see the Aharonov–Bohm effect at work and link this to coherent domains [43]. A promising theoretical background offering a fresh understanding of the UHD signal transfer phenomenon may reside in the so-called Zhadin’s effect when coupled with the ion cyclotron resonance (ICR) theory, started by Liboff [44,45] in the framework of bioelectromagnetics and followed by Zhadin and the group of Italian physicists (Preparata, del Giudice, Giuliani, etc. [46,47]), who have deepened and refined the theory by rooting it in QED. It is only in this context that both the ICR and the Zhadin effect have been shown to be fully scientifically explainable. With some further experiments and considerations, it became possible to scientifically understand and explain UHD phenomena, either pertaining to the same bulk of liquid (as we did in [1]) or when it is about the UHD signal transfer.

In the first of the two papers, we have already confirmed the impact of the UHD signal of a-IFNγ transfer from one solution to another via glass [1], even if we dealt primarily with the physicochemical and UV/VIS spectroscopical identification of the UHD in highly diluted solutions (at least 10^−18^ of the original solution), and their further processing through dilution accompanied by vigorous shaking (10^−2^ dilution of the original). In the present paper, we focus on further research regarding different UHD signal transfer modalities.

As already said, for the biologically active compound that was subject to ultra-high dilution processes, we chose antibodies to IFNγ (marked as a-IFNγ) since the behavior of this specific active biological substance (and also a-IFNγ itself) has already been subject to extensive research with a noticeable outcome [2,48,49,50,51]. As for UHD solutions, no additional safeguards (like shielding from electrosmog) were applied in the transfer studies. This way, they were comparable regarding the strength and stability of the UHD signal.

Given the pilot phase of the study, we have tried to answer, at least roughly, different questions through various experimental situations. In the previously published paper [1], we have shown that the effect (nature) of the UHD signal in the transfer is preserved with respect to the original one in the UHD solutions and that, in general, it proved to be weaker. We also confirmed that usually, the signal from a-IFNγ is stronger than the signal from distilled water. We also keep the same expectation for the UHD transfer research presented here. In this paper, we will present the results of experiments we have carried out to investigate some further characteristics of non-contact UHD signal transfer.

We have tried to answer the question of whether the signal is better transmitted through the glass than through the air (plus glass that still borders the liquid). We expected that it would be better transferred through the glass.Related to the previous question was the second one, namely, whether the signal is better transmitted at 1 cm than at 4 cm distance between the donor and the receiver. Following the known working of various fields in nature, we expected that the detection of the UHD signal with the solutions exposed at a 4 cm distance would be weaker than with the 1 cm exposure since a longer distance would reduce the transfer of the signal.We also tested some further variations of the exposure time and the stability of the signal after physically separating the donor from the receiver. Here, we expected that after 1 h of separation, the effects of UHD signal transfer would be weaker than before separation.The fourth question related to activation as a necessary component of UHD signal transmission. We expected that activation is necessary for signal transfer but that perhaps the signal can still be transmitted without activation if the receiving solution is exposed to the donor for a prolonged period.

We would also like to point out that, unlike the research we published in our first paper [1], in this paper, we are not so concerned with the type of signal (two different a-IFNγ UHD solutions). At this pilot stage of the signal transfer investigation, we are primarily focused on the difference between the signal of the active biological substance and that of the control (obtained from distilled water).

## 2. Results

### 2.1. Transfer via Glass (0 cm) with Activation through Different Times of Exposure and Separation

Table 1 below gives Cohen’s D values of different measurements of different UHD signal transfers regarding different exposure times, spanning from one hour to overnight exposure. In general, we may observe that the effects are weak immediately after activation (time 00), becoming more pronounced after one hour of separation (time 01). Very noticeable are the effects after 1 h of exposure (time 10), somewhat fading after 1 h of separation (time 11). The effects appear most conspicuous after overnight exposure (time N0).

As can be seen in Table 1, most of the experiments were performed with an a-IFNγ mix. The table also shows that the transfer effects are stronger with longer exposures after activation. In Figure 1, we present the difference between the UHD signal and that of the control water, as detected by the ORP measurement method. We can notice that the statistical significance observed after one hour of exposure (time 10) was lost after separation (time 11).

If separation caused the signal to fade, a longer exposure (overnight, see Figure 2, ORP measurements), as expected, enhanced the signal expression.

Regarding overnight exposure and UV spectroscopy, in addition to the result quoted in Table 1, we also performed a statistical analysis of the raw data at 195 nm wavelength. Here, we observe a high statistical significance (Figure 3). It is also worth noting that after separation (time N1), the signal roughly retained size (Cohen’s D) and even increased significance (*p*).

### 2.2. UHD Signal Transfer over Distance (via Air) with Activation

Table 2 presents the results of experiments where we investigated a possible UHD signal transfer through the air (or vacuum behind the air). The distance concerns the separation between the donor bottle and the receiver bottle. The results are presented in Cohen’s D values to be directly comparable with other results. We may observe considerable similarities in effects. It is worth noting that after 1 h of separation between donor and receiver (time 01), the effects almost vanish, while this is not the case with the transfer via glass.

The two figures below (Figure 4a,b) show the differences between the distances at time 00 immediately after activation. For the distance 0 cm, we present a-IFNγC9-sig. for comparison, as it showed higher effect sizes than the a-IFNγ mix-sig. At a distance of 1 cm, we experimented only with the a-IFNγ mix-sig., and at 4 cm with the a-IFNγC9-sig. With conductivity measurements of the signal transfer via 1 cm, we do not notice a significant statistical difference between the UHD signal and the control. However, the signal appeared stronger at 4 cm. With ORP measurements at a distance of 0 cm, no difference is seen between the UHD signal and the control. The transfer over distance demonstrates statistically significant differences in ORP. Here, the difference is higher at 1 cm than at 4 cm.

### 2.3. Experiments without Activation

Stemming from general research experiences, we estimated activation as a significant factor in the UHD signal transfer. Therefore, we expected that no signal would be transferred, at least with short exposure times. With longer exposure times, we expected a possibility of some noticeable signal transfer.

Table 3 below presents the results of physicochemical measurements of the a-IFNγC9 signal at two different exposure times (one hour (time 10) or overnight (time N0)) and after separation (times 11 and N1). As seen from Table 3, the effects were rather weak after only one hour of exposure, with nothing significant appearing after one additional hour of separation. Overnight exposure yielded quite conspicuous effects, especially in pH. It is worth noting that the latter parameter changed its sign after the additional hour of separation. The results of UV spectroscopy of the a-IFNγ mix one hour after exposure (time 01) and after separation (time 11) are also presented; however, here, we do not observe any significant differences between the a-IFNγ mix and the control.

Figure 5 displays the results of pH measurements in overnight experiments. After separation, the pH reverses while the statistically significant difference increases.

## 3. Discussion

In general, via our measurement system, we detected the transfer of the UHD signal of a-IFNγ solutions in all tested situations. The detection thus corroborates similar reported findings from other groups [2,35,37,38,39,40,41,51,52,53,54]. Even the solvent (water) UHD signal—presumably stemming only from striking—was detected, although the detection was much weaker than the one produced by the other two signals. Therefore, our basic expectation was corroborated.

In the second part of the research, we tried to answer four questions posed in the Introduction. The first of them tackled the expectation that the signal would be better transmitted through the glass than (additionally) via air. As seen in Table 2 and Figure 6, the assumption has not been validated. Namely, the overall effects (disregarding the sign of Cohen’s D, Table 2) at time 0 are more visible over distances (1 or 4 cm) than if the transfer occurred via glass (0 cm). Even if we focus only on the electrical conductivity at time 0, we see very similar values. However, we cannot yet draw a definitive estimate, as some effects may only appear after a certain time. It is evident in the detection of the pH change in the transfer via glass, as can be seen in the experiment without activation (see Table 3). In the second expectation, we assumed that the UHD signal transfer would be weaker at 4 cm than at 1 cm. The results show that increasing the distance (0 cm–1 cm–4 cm) between the donor and the receiver increases the standard effect size of the transmission very slightly (insignificantly) when detecting the UHD signal of a-IFNγ solutions (see Table 2 and Figure 6). These pilot experiments regarding the UHD signal transfer over shorter distances, with their somewhat surprising results, particularly call for much more systematic research. Such research, in addition to investigating greater distances, should also encompass different timeframes, where delayed effects might also come to the fore.

In general, the third assumption posed in the Introduction that longer exposure times will result in a stronger signal is also validated, at least by summarizing the physicochemical detections. It is most clearly illustrated in Figure 7 (see also Table 1), where one may see that in the summation of the absolute Cohen’s D values in physicochemical measurements, the strongest effects can be noticed in overnight exposure. The effects here are the lowest at 00 exposure time experiments and steadily growing through longer exposures. Interestingly, in UV/VIS spectroscopy, the highest effect is observed after one hour of separation (01), after which it steadily decreases (see the violet columns in Figure 7). It indicates that from the signals captured in the receiver solutions, we detect different aspects by different measuring methods. It is interesting to note that contrary to these transfers via glass experiments, in the experiments over distance, the separation resulted in a significant lessening of the signal, especially in conductivity measurements, thus corroborating the assumption (see Figure 8 and Table 2).

Hypothesis four, posed in the Introduction, namely that the activation by striking will enhance the transfer of signal and thereby its detection in the receiver solution, was also corroborated by experiments. If we consider Table 3 for the identical exposure/separation times, we see much stronger effects in the case of activation than in the case of its absence. More conspicuous effects can be observed only in the overnight experiment, but even here, a reversal of the effect (in pH) after one hour of separation occurred. Furthermore, as displayed in Figure 9, the trend intensifies over the exposure duration, culminating in the most pronounced impact during the overnight experiment. However, notably, a reversal in the effect begins to manifest just one hour after the separation process.

More experiments in this area should be performed to assess the nature of the signal transfer due only to exposure without activation.

Regarding the theoretical background that can shed light on the UHD signal transfer phenomena presented here, due to its theoretical depth and many empirical confirmations, we decided on the Zhadin effect, which is related to the ICR theory taken in the context of QED. The latter defines the environmental electromagnetic conditions of the static magnetic field and the much less intensive AC field in which fast ion currents (pulses) are triggered [55]. The currents represent ions, which, via resonance conditions, receive enough energy to escape their proper coherent domains (see also [56]). The currents induce a magnetic field that spreads in the surroundings and can thus work as a signal passing through glass and, of course, through air. It then induces electrochemical phenomena in an otherwise chemically separated liquid.

Of particular importance in this respect is the discovery that the hydronium ion (H_3_O^+^) also undergoes the same ICR mechanism in relation to coherent domains and that under ICR conditions of ambient DC and AC magnetic fields, there is a rapid increase in conductivity in water, which then slowly decreases [57]. The highly significant pH change in the unactivated exposed liquid (Table 3, Figure 5) may reflect the action of the ambient fields on the hydronium ion. The action on the hydronium ion could also explain the mostly increased values of the electrical conductivity in the liquids exposed to the UHD signal (see Table 1 and Table 2).

## 4. Materials and Methods

### 4.1. General

There is a large body of research on scientifically not yet universally accepted and theoretically well-founded phenomena in water, with a variety of detection and measurement methods, but no generally accepted methodology yet. Moreover, there is no specific sensor for dynamically ordered states of mesoscopic water. In this case, one has to resort exclusively to a statistically based measurement system. In order to ensure the highest possible reliability of the measurements, the water research team of the BION Institute has devoted special efforts to the development and calibration of physicochemical and physical systems. They include the associated protocols, which must be both robust and sensitive enough to detect the subtle phenomena under investigation. The system developed so far and used in the present study includes physicochemical methods (pH, ORP, and conductivity) and UV/VIS.

### 4.2. Liquids

#### 4.2.1. Original (Received) Waters and Solutions

All original liquids (water and solutions, four different liquids altogether) subject to research were obtained from OOO “NPF ”Materia Medica Holding”, Moscow, Russia (hence designated as MMH). They were the unprocessed (no dilution or shaking) distilled water that played the role of control in experiments. These waters were used as donors for the UHD signal. These original liquids are as follows.

Distilled water used for further dilution of biologically active substances (W).

The same (W) water processed as if diluted in itself by nine centesimal sequential dilutions (9 × 10^−2^) and sequentially shaken between each “dilution” will be named WC9.

The third original liquid, prepared by MMH, was a processed dilution of antibodies to interferon-gamma (a-IFNγ). It was processed in the same ninefold centesimal manner as the second liquid (9 × 10^−2^) and will be named a-IFNγC9. Here, we followed the finding of Ryzhkina et al. [48], who performed a systematic study of different potencies involving the same starting substance (mother tincture) and found by far the highest effect (electrical conductivity and pH) with C9.

The fourth liquid was a multiple-processed dilution of IFNγ. The liquid represented the following mixture of successive centesimal dilutions (10^−2^ designated as C1) with intermediate shaking: C12, C30, and C50 (a-IFNγ C12C30C50) and will be named a-IFNγ mix. Regarding the decision for this specific mixture, we followed the finding of Don et al. [58], who conducted a study with the same combination of potencies and the same starting substance and found a statistically significant effect of the UHD signal.

All samples were supplied in the polypropylene flasks, but in advance of the experiment, were poured from the polypropylene to the glass flasks (Duran^®^ GL 45, UAB Santonika, Kaunas, Lithuania), protected from direct intense light, and stored at room temperature with closed lids (20–23 °C). The reason behind the choice of Duran glass was the fact that some other studies of similar phenomena used glass where the transmission stopped somewhere around 300 nm (see, for instance, [2]). However, we know that our detection may, therefore, have been limited because some signals could be transmitted at shorter UV wavelengths.

The samples were stored in places with relatively low EM pollution. Care was taken to ensure that all samples were subjected to very similar (if somewhat variable) ambient EM noise. Its maximal value throughout measurements was registered at the average frequency of 50 ± 1.5 Hz; its average value of magnetic flux intensity was 22.6 ± 3 nT. This ELF field was almost the same in all directions. We also measured the static (geomagnetic) magnetic field in the laboratory rooms, whose flux density (B) values were horizontal vector (H) = 23 µT, vertical vector (V) = 21 µT and total 3D vector 31 ± 3 µT. 

#### 4.2.2. UHD Signals Receiver Solution

For investigating the UHD transfer characteristics, we used a specially developed solution with the following composition: 0.413 mL of 3% hydrogen peroxide (Lekarna Ljubljana Pharmacy, Ljubljana, Slovenia) per liter (12.39 mg/L of pure H_2_O_2_; i.e., 0.36 mM H_2_O_2_) and 0.2 g/L sodium hydrogen carbonate (Solvay Chimica Italia S.p.A, Rosignano, Italy; 0.01 M NaHCO_3_) diluted on distilled water provided by the Bion distiller (χ = 2 µS, Kambic laboratory equipment d.o.o., 8333 Semic, Slovenia). We decided on this receiver solution following our previous experiences and additional examination of optimizing possible liquids for the present investigation, as well as taking into consideration the profound study by Voeikov [59]; see also [1].

### 4.3. Physical Transfer of UHD Signal (via Glass Contact, via Air)

#### 4.3.1. General

Principally, we performed the transfer of the UHD signal of the original substances (donor) via glass or air at distances 0, 1, and 4 cm by applying strokes to the bottle with the donor liquid in the direction of the bottle containing the receiver solution. This method is called “with activation”. It means that the glass bottle with the donor substance was stricken 15 times with a mallet shaker (striking). The estimated force exerted during the manual strokes is approximately 0.04 N, with a frequency of 2 Hz. The activation was performed via glass (directly), where the donor bottle was touching the receiver bottle (0 cm), and through the air, where the donor bottle was 1 or 4 cm away from the receiver bottle. While doing the air transfer through striking, the donor bottle had no contact with the surface on which the receiver liquid bottle was placed. We also conducted a pilot study “without activation” where we did not strike the donors; however, the donor bottle was touching the receiver bottle (0 cm) at different exposure times. We mark the transferred signals from the original substances as W-sig. (from W), WC9-sig. (from WC9), a-IFNγC9-sig. (from a-IFNγC9) and a-IFNγ mix-sig. (from the a-IFNγ mix).

#### 4.3.2. Distances between the Donor and Receiver

The activation was performed via glass; the following measurements were conducted:

- immediately after activation at a distance of 1 cm (mark 00) and one hour after separation (mark 01),

- immediately after activation at a distance of 4 cm (mark 00) and one hour after separation (mark 01).

#### 4.3.3. Exposure and Separation Times with Activation

We studied different times of exposure of the receiver solution to the UHD signal. In addition, we also studied the reception and expression of the UHD signal after different exposure times and a one-hour separation of the donor and receiver bottles by moving the previously attached bottles away from each other for 2 m. The purpose of this experiment was to repeat the measurement of the solutions’ parameters after one hour of separation of the bottles. In all these experiments, activation was performed at 0 cm. The situations were as follows (for the schematic design (flow charts), see also Figure 10 and Figure 11):Immediately after activation (mark: 00) and in continuation.One hour after separation (separation, mark: 01).One hour of exposure after activation (exposure, mark: 10) and in continuation.One additional hour after separation (exposure + separation, mark: 11).Overnight exposure (long exposure, mark: N0) and in continuation.One additional hour after separation (long exposure + separation, mark: N1).

**Figure 10 ijms-25-00656-f010:**
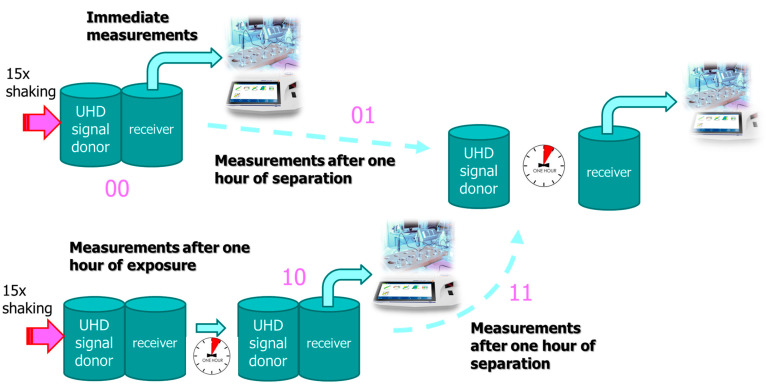
Flow charts of experimental situations for different types of experiments lasting up to two hours.

**Figure 11 ijms-25-00656-f011:**
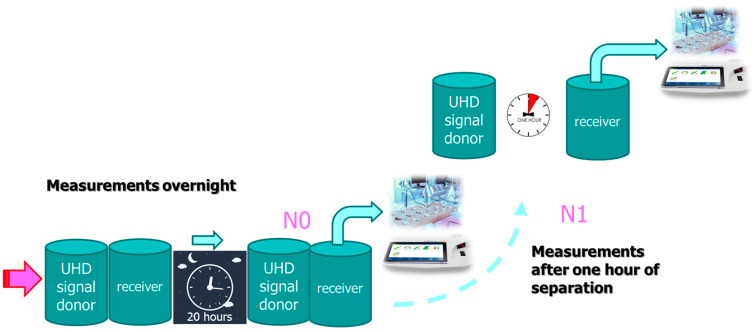
Flow charts of experimental situations for overnight experiments.

#### 4.3.4. Transfer without Activation

Without activation (striking), all experiments were performed at a distance of 0 cm (via glass) with different exposure times:One hour of exposure after activation (exposure, mark: 10) and in continuation.One additional hour after separation (exposure + separation, mark: 11).Overnight exposure (long exposure, mark: N0) and in continuation.One additional hour after separation (long exposure plus separation, mark: N1).

### 4.4. Measurement Methods

#### 4.4.1. General

For measurement of the physicochemical parameters, we used The Vernier Go Direct^®^ devices (Vernier Software & Technology, Beaverton, OR, USA): Temperature Probe, pH Sensor, Conductivity Probe, and ORP Sensor. We employed simultaneous temperature measurements as a control to ensure that the observed data differences did not result from temperature variations. Conductivity is known to have a strong correlation with temperature, and to a lesser extent, this also pertains to the other measurement methods used. Each individual measurement was conducted in a separate beaker. The measured range of accuracy (which deviates from the officially declared value and is not part of the observed, investigated, and considered measurement drift) for these three devices is as follows: 0.5 µS/cm for conductivity, 2 mV for ORP, and 0.02 for pH.

For UV/VIS absorption spectroscopy measurements, we used a Macherey–Nagel spectrophotometer (Düren, Germany), wavelength range 190–1100 nm, with a 50 mm quartz cuvette cell. The measured accuracy range here is 0.003. Here, too, each measurement was performed in a single glass beaker.

#### 4.4.2. Measurement Protocol

For the physicochemical measurements of the original substances, the liquids obtained in the bottles were poured into beakers. All physicochemical parameters (conductivity, ORP, pH) and temperature were measured simultaneously by two sets of measuring devices. In order to prevent possible slight measurement errors due to the drift of measurements over time, we did not measure different solutions consecutively but alternately. Physicochemical methods such as conductivity and pH have also been used in various other studies to detect signals in ultra-high diluted solutions [60,61,62].

For UV/VIS absorption measurements, the receiver solution was poured into a 50 mm quartz cuvette cell, wherein measurements were performed alternately with different samples. All the measurements were relative to the reference (W-sig.) that was not shaken. Due to possible slight measurement drift, we performed the measurements according to the procedure explained for the physicochemical measurements above.

Due to a larger variation in the UV absorption with the physical transfer of the UHD signal, we decided not to choose a particular wavelength but the average absorption values of the band spanning from 190–210 nm.

### 4.5. Experimental Plot

#### 4.5.1. Physicochemical and UV/VIS Spectroscopy Experiments with Different Distances between the Donor and Receiver

In the second set of experiments, the UHD signal transfers of the a-IFNγ mix and a-IFNγC9 donor substances were tested in relation to different distances (1 and 4 cm) between donor and receiver solutions (bottles) at different times of exposure after activation and an additional hour after separation (see Table 4).

#### 4.5.2. Physicochemical and UV/VIS Spectroscopy Experiments with Different Exposure and Separation Times

In the first set of experiments, the UHD signal transfer of the donor substances was tested in relation to different times of exposure and separation. The transfer was implemented via activation at 0 cm between the donor and the receiver solutions. The experiments were performed with different original solutions; see Table 5 for more detail.

#### 4.5.3. Physicochemical and UV/VIS Spectroscopy Experiments in the UHD Signal Transfer without Activation

To check if the molecular information transfer can also be achieved without activation, i.e., just via spontaneous emission of the subtle field of the UHD signal stored in the donor solution, we performed experiments using the UHD signals of the a-IFNγ mix and a-IFNγC9 at different exposure times: 1 h (10 and 11, see Section 4.3.4 for more detailed explanation) and overnight (N0, N1), and the distance 0 cm (see Table 6).

### 4.6. Statistical Analysis of the Results

For estimating statistical significance, we applied appropriate tests regarding normality and the number of groups. In the case of the normal distribution, we used ANOVA; otherwise, we used the Friedman test. For post hoc analysis, we subjected the groups to a post hoc *t*-test (normal distribution) or/and Wilcoxon signed-rank test. All tests were performed via pairwise comparison.

Statistical data analysis was performed using XLSTAT statistical software (XLSTAT PREMIUM-Evaluation 2022.3.1) for Excel (2021). For basic statistical parameters of groups, we calculated the average, standard deviation, standard error, and normality with the Shapiro–Wilcoxon test. We consider that the pairwise method of the Shapiro–Wilcoxon test is appropriate for two reasons: (a) the samples are not unrelated, as they have the same original distilled water, and (b) there is often a slight drift in the measurements, which can lead to a systematic measurement error in the required precision range in detecting statistically significant differences. To estimate statistical significance in the data variation, we used Levene’s test or F-test for differences in variance and Cohen’s D for assessing standardized effect size.

Differences are considered statistically significant with *p* < 0.05 or if 0.1 < *p* < 0.05, while the absolute Cohen’s D value is higher than 0.5.

To make various comparisons, not only between the substances in their original or processed forms but also between the efficacy of the used physicochemical methods and the methods of UHD signal treatment, we resorted to the calculation of the average of absolute Cohen’s D values, where the latter were calculated vs. control (W). Thus, we could have performed synthetic evaluations of the study results from different perspectives.

## 5. Conclusions

In conclusion, we can say that in this pilot study, we have also been able to confirm the existence of signal transfer based on an ultra-dilute biologically active compound, in our case, an antibody to IFNγ. Although the detection of the physically transferred signal is generally weaker than the signal in the original UHD solutions, it is still detectable with our measurement system and protocol, both in terms of effect (Cohen’s D) and statistically significant differences. In this respect, different detection methods may detect the signal differently (see Figure 7 and Table 1, Table 2 and Table 3).

Our second expectation, expressed in the Introduction, that the distance between the donor and the receiver would reduce the effect of the transferred signal, was not confirmed in our experimental model, as on average, we obtained similar results with increasing distance from 0 to 4 cm (see Figure 6). Noticeably, however, all three distances have a weaker effect than the original UHD solution (see Figure 6). In further experiments, we propose to increase the distance between the donor and the receiver.

Regarding different exposure times or further separation, the study showed a differing outcome with physicochemical measurements compared to the UV/VIS ones. With UV/VIS measurements, we see a high rise of Cohen’s D from time 00 to times concerning one hour of separation (01) or one hour of exposure (11); however, with a longer time (11 or N1), the effect lessens (Figure 7). On the other hand, with physicochemical measurements, a longer exposure time of the signal consistently shows an increasing effect (Figure 7). However, the conductivity measurements at the three different distances consistently show the vanishing of the UHD signal after one hour of separation (Figure 8). More experiments with more time variations should be conducted in the future.

As expected, activation was shown to be an important factor for the UHD signal transfer, at least for shorter times. If we compare the results for the physicochemical measurements between Table 1 (activation) and Table 3 (no activation) at the same shorter times (10, 11), it can be seen that the results of the activation measurements exceed the results of the no activation measurements by many times. However, the effects of overnight exposure (averages of absolute Cohen’s D, time N0) are already very similar. As hypothesized, longer exposure times may compensate for activation alone. More systematic experiments should be performed in the future to see when and how this compensation comes into force.

An important outcome of the experiments is the finding that shorter exposure times, even with the glass–glass contact, do not necessarily transfer the UHD signal from one fluid to another, which can be important for developing a more convenient research methodology regarding the placement of control samples. It is because they are mostly installed far away from UHD samples, so as not to be “contaminated” by the UHD signal. However, the conditions where control samples are far from the UHD ones may bring other drawbacks, for instance, a different temperature or a changed electromagnetic environment. These factors may then influence these sensitive experiments. If—as indicated in the present study—control samples may be closer to the UHD ones, we would have more comparable samples.

From a theoretical point of view, it seems most justified that the mechanism behind the UHD signal transfer is the ICR mechanism, as considered by the current QED in relation to coherent water domains. Future experiments of this kind could, therefore, follow the transfer of UHD signals under artificially altered electromagnetic conditions in the experimental environment in a more targeted way, for example, testing the possible transfer of a UHD signal coupled to a hydronium ion.

## Figures and Tables

**Figure 1 ijms-25-00656-f001:**
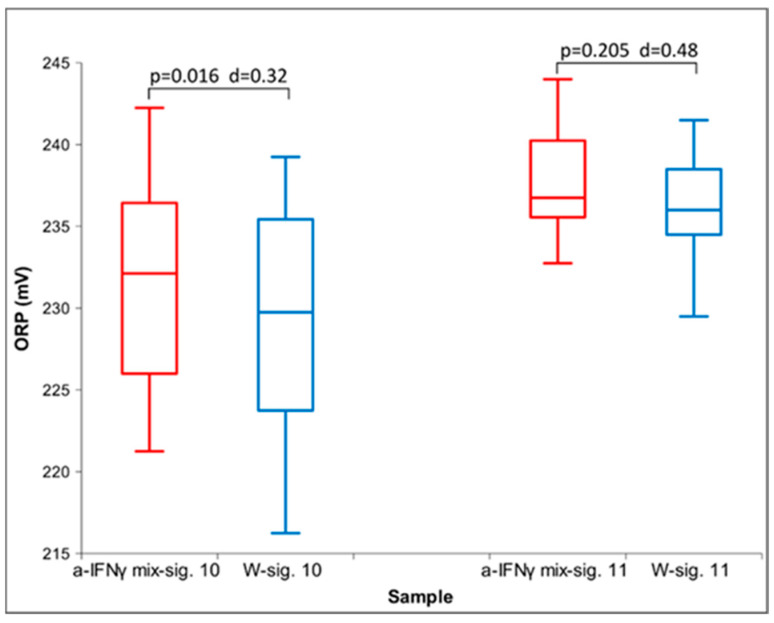
Box plot of ORP measurements presenting median and quartiles. The box plots in red show the values for the UHD signal, and the box plots in blue for the control. One-hour exposure (time 10), and one hour after separation (time 11) (*n* = 16). *p*-values and Cohen’s D are presented, too.

**Figure 2 ijms-25-00656-f002:**
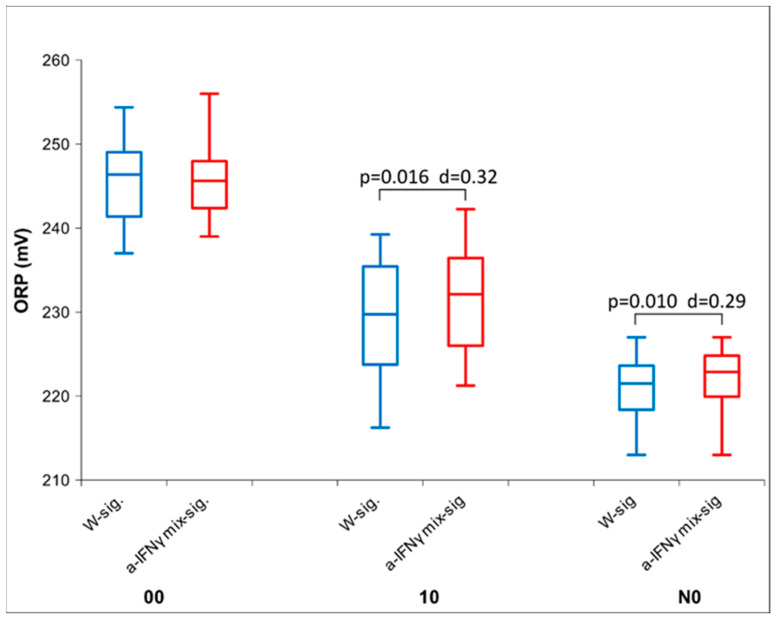
Box plot of ORP measurements presenting median and quartiles at 0 cm with different times of exposure (immediately after activation (00), after one hour of exposure (time 10), and overnight (time N0) (*n* = 16). The box plots in red show the values for the UHD signal, and the box plots in blue for the control; *p*-values and Cohen’s D are presented, too.

**Figure 3 ijms-25-00656-f003:**
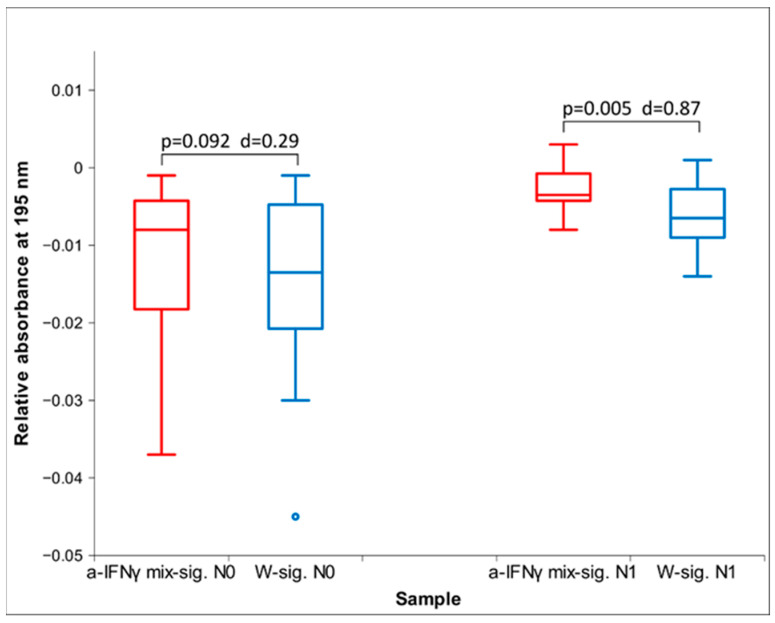
Box plot of relative absorbance at the waveband 195 nm at 0 cm after overnight exposure (time N0) and one hour after separation (time N1) (*n* = 12). The box plots in red show the values for the UHD signal, and the box plots in blue for the control; *p*-values and Cohen’s D are presented, too. The tiny circle represents an outlier.

**Figure 4 ijms-25-00656-f004:**
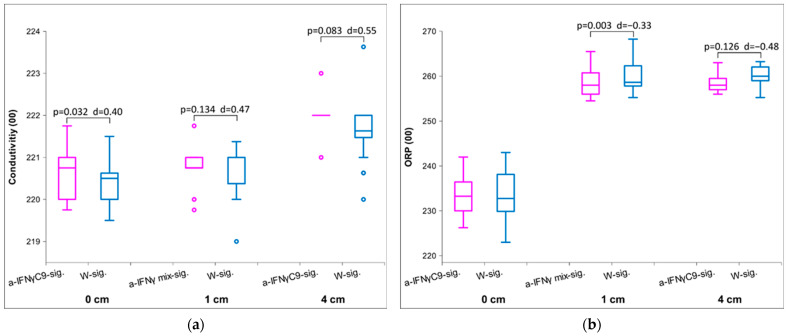
(**a**) Box plot of measurements in conductivity presenting median and quartiles immediately after activation at different distances of exposure (*n* = 16); (**b**) Box plot of ORP measurements presenting median and quartiles immediately after activation at different distances of exposure (*n* = 16). The box plots in pink show the values for the UHD signal, and the box plots in blue for the control; *p*-values and Cohen’s D are presented, too. The tiny circles represent outliers.

**Figure 5 ijms-25-00656-f005:**
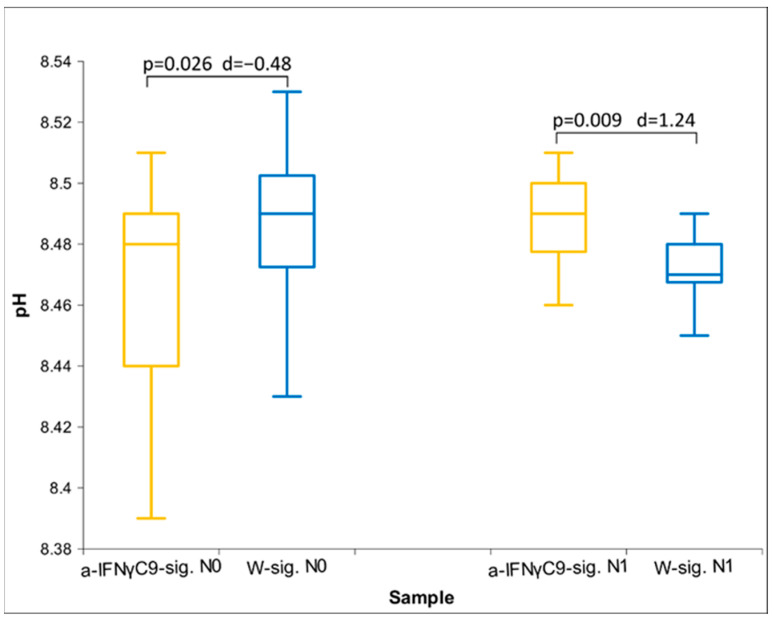
Box plot of ORP measurements presenting median and quartiles of pH with different times of exposure (*n* = 16). The box plots in yellow show the values for the UHD signal, and the box plots in blue for the control; *p*-values and Cohen’s D are presented, too.

**Figure 6 ijms-25-00656-f006:**
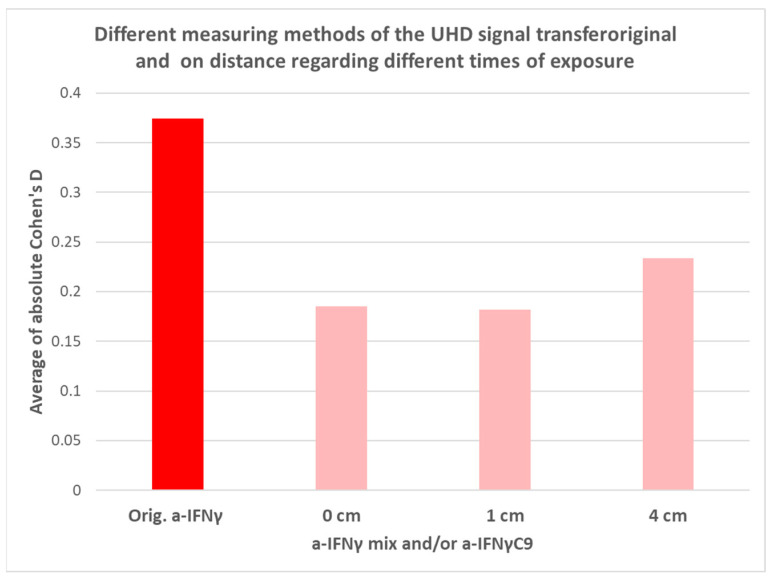
An illustration of the effect of transmitting a UHD signal over different distances, as measured by the average of the absolute Cohen’s D values for all three physicochemical measurements. The red column (leftmost) shows the standard effect of the original solution (vs. its control), while the other columns (pink) show the effects at different distances between the donor and receiver. Source substance: 0 cm: a-IFNγC9-sig., 1 cm: a-IFNγ mix-sig. and at 4 cm: a-IFNγC9-sig.

**Figure 7 ijms-25-00656-f007:**
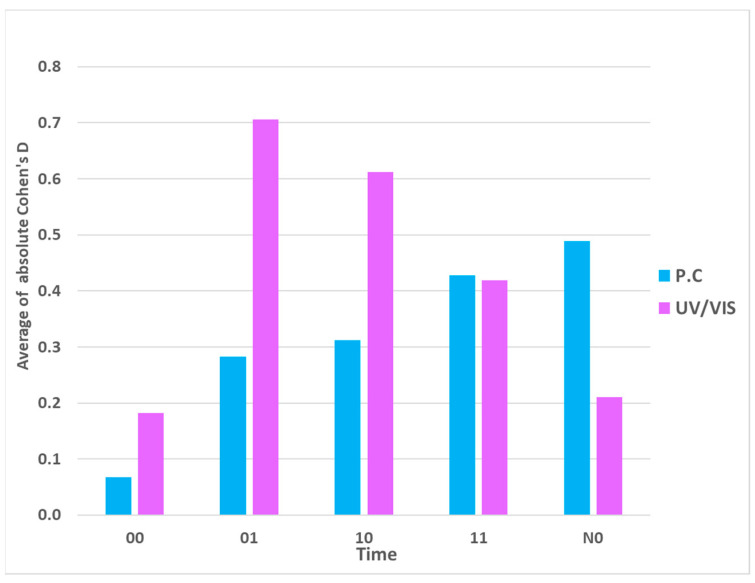
Graph showing the effect of the UHD signal transfer (a-IFNγ mix, a-IFNγC9) registered by physicochemical and UV/VIS spectroscopy measurements. In this experimental study, the following physicochemical measurements were performed conductivity, oxidation–reduction potential (ORP), and pH. The effect of the UHD signal transfer of a-IFNγ mix via the average of the absolute Cohen’s D values for different times is shown. The time labels: immediate measuring after activation—00, one additional hour of separation—01, one hour of exposure after activation—10, after an additional hour of separation—11, overnight—N0.

**Figure 8 ijms-25-00656-f008:**
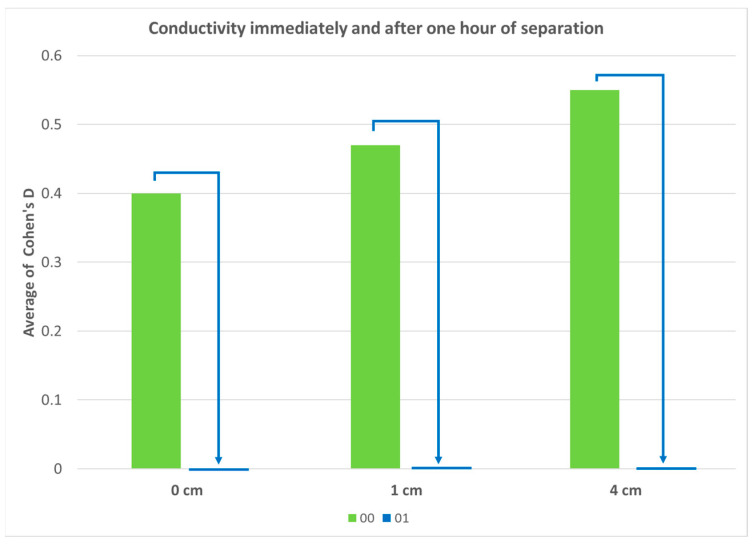
Graph showing the effect of the UHD signal transfer (a-IFNγ mix or a-IFNγC9) recorded by conductivity measurements. The Cohen’s D values for different distances and times are shown. The time labels: immediate measuring after activation—00, one additional hour of separation—01.

**Figure 9 ijms-25-00656-f009:**
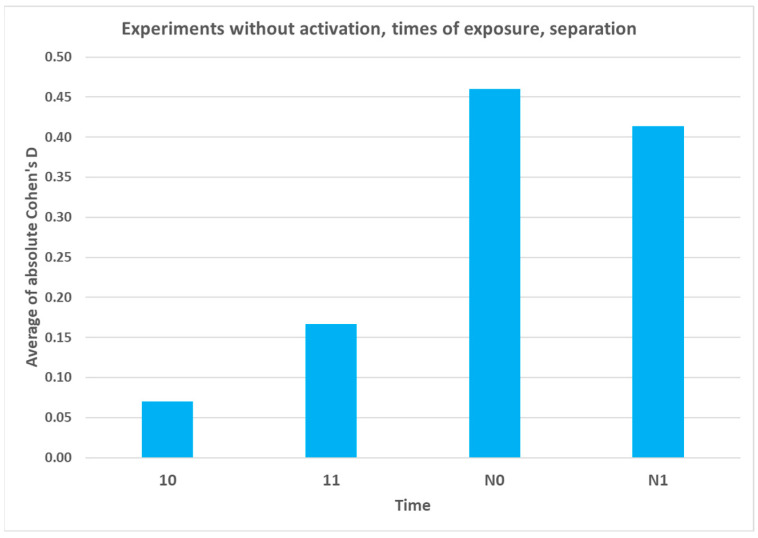
Graph showing the effect of the UHD signal (a-IFNγC9) transfer without activation recorded by physicochemical measurements. The average absolute Cohen’s D values for different distances and times are shown. The time labels: one hour of exposure after activation—10, after an additional hour of separation—11, overnight—N0 and after an additional hour of separation—N1.

**Table 1 ijms-25-00656-t001:** Cohen’s D values belonging to different measuring methods of the UHD signal transfer with activation via glass, regarding different times of exposure (immediate measuring after activation—00, one additional hour of separation—01, one hour of exposure after activation—10, after an additional hour of separation—11, overnight—N0 and after an additional hour of separation—N1). Values matching the statistical significance (*p* < 0.05) are indicated in bold. The gray zone designates that these experiments were not conducted.

		Times					
UHD Signal	Cohen’s D	00	01	10	11	N0	N1
a-IFNγ mix	χ	0	−0.18	0	−0.8	**1.18**	
	ORP	0	**−0.59**	**0.32**	0.48	**0.29**	
	pH	0.05	0.39	**0.62**	0	0.25	
	UV	**0.27**	**1.05**	0.14	0.33	0.21	0.26
a-IFNγC9	χ	**0.4**	0				
	ORP	−0.05	0				
	pH	0.23	**0.71**				
	UV	0.09	0.36	**1.08**	0.51		
WC9	χ	0.17	**−0.58**				
	ORP	0.09	−0.31				
	pH	−0.15	0.61				
	UV	−0.07	0.01	0.27	0.19		

**Table 2 ijms-25-00656-t002:** Cohen’s D values of different measuring methods of the UHD signal transfer (a-IFNγC9-sig. at 0 cm, a-IFNγ mix-sig. at 1 cm, and a-IFNγC9-sig. at 4 cm) on distance regarding different times of exposure. Values matching statistical significance (*p* < 0.05) are marked in bold.

Distance	0 cm		1 cm		4 cm	
Method/Time Variations	**00**	**01**	**00**	**01**	**00**	**01**
χ	**0.40**	0	**0.47**	0	**0.55**	0
ORP	0	0	**−0.33**	−0.29	−0.48	0
pH	0	**0.71**	0	0	0	0.37

**Table 3 ijms-25-00656-t003:** Cohen’s D values of different measuring methods of the UHD signal transfer without activation with different times of exposure (one hour of exposure—10, after an additional hour of separation—11, overnight—N0 and after an additional hour of separation—N1). Values matching the statistical significance (*p* < 0.05) are indicated in bold. The gray zone indicates that these experiments were not conducted.

Experiments without Activation					
		Times of Exposure, Separation			
UHD Signal	Methods	10	11	N0	N1
a-IFNγC9	χ	0	0.5	−0.50	0
	ORP	0	0	0.40	0
	pH	0.21	0	**−0.48**	**1.24**
a-IFNγ mix	UV	0	0		

**Table 4 ijms-25-00656-t004:** Physicochemical measurements and UV spectroscopy with a-IFNγ mix and a-IFNγC9 substances at a distance of 1 and 4 cm at different times.

	1 cm				4 cm	
	00	01	10	11	00	01
Conductivity	a-IFNγ mix	a-IFNγ mix	-	-	a-IFNγC9	a-IFNγC9
ORP	a-IFNγ mix	a-IFNγ mix	-	-	a-IFNγC9	a-IFNγC9
pH	a-IFNγ mix	a-IFNγ mix	-	-	a-IFNγC9	a-IFNγC9
UV	a-IFNγ mix	-	a-IFNγ mix	a-IFNγ mix	-	-

**Table 5 ijms-25-00656-t005:** Physicochemical measurements and UV spectroscopy with different substances at a distance of 0 cm at different times of exposure and after separation.

	00	01	10	11	N0	N1
Conductivity	All 3	All 3	a-IFNγ mix	a-IFNγ mix	a-IFNγ mix	-
ORP	All 3	All 3	a-IFNγ mix	a-IFNγ mix	a-IFNγ mix	-
pH	All 3	All 3	a-IFNγ mix	a-IFNγ mix	a-IFNγ mix	-
UV	All 3	All 3	All 3	All 3	a-IFNγ mix	a-IFNγ mix

**Table 6 ijms-25-00656-t006:** Physicochemical measurements and UV spectroscopy with a-IFNγ mix and a-IFNγC9 substances at a distance of 0 cm at different times, without activation.

	10	11	N0	N1
Conductivity	a-IFNγC9	a-IFNγC9	a-IFNγC9	a-IFNγC9
ORP	a-IFNγC9	a-IFNγC9	a-IFNγC9	a-IFNγC9
pH	a-IFNγC9	a-IFNγC9	a-IFNγC9	a-IFNγC9
UV	a-IFNγ mix	a-IFNγ mix	-	-

## Data Availability

Research data are available from the BION Institute upon special request.

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
