# Peer review of "Molecular Signal Transfer of Highly Diluted Antibodies to Interferon-Gamma Regarding Kind, Time, and Distance of Exposition"

_ijms, 2024, doi:10.3390/ijms25010656_

Round 1

Reviewer 1 Report

Comments and Suggestions for Authors

The manuscript "Molecular signal transfer of highly diluted antibodies to interferon-gamma regarding kind, time and distance of exposition" describes the possibility of transfer of the ultra-high dilutions (UHD) signal from the UHD solutions to the receiver solution, in particular, if the molecular source used in the donor solution was the biologically active antibodies to interferon-gamma molecule. The main aim was to determine how the transfer of the UHD signal is affected by the time of exposure of the receiver to the donor, the distance between the two, and how the transfer is affected by activation (shaking) versus exposure alone.

The work is interesting, and the result could be judged to be worth publishing in this journal. However, some revisions should be made to improve the scientific quality of the results mentioned in this manuscript.

Additional comments that need to be addressed:

1)     Please explain how you make activation by shaking. I would like to know the details because nowhere I could not find it.

2)     The authors claimed: "The activation was performed via glass; the following measurements were conducted:" the sentence is unclear. Does it mean that diluted solutions were in bottles?

3)     Please explain the abbreviations a-IFNγ, a-IFNγC9, and WC9.

4)     What do the authors mean by physicochemical measurements (Figure 7)? Please explain this phenomenon in depth.

5)     The introduction is pretty too long. I suggest to shorten it.

Comments on the Quality of English Language

Although the manuscript is written well enough in English to be understood, the English language and style (grammar) must be carefully revised throughout the whole manuscript. 

Author Response

Please see the submitted review.

Reviewer 2 Report

Comments and Suggestions for Authors

Abstract:

- Information about how the anti-gamma-IFN signals were evaluated must be included. - The keywords are too specific and excessive. They must be restricted to up to five and comprehensive enough to be found in the main scientific databases.   Introduction:   - Second paragraph - The sentence "This investigation employed three physicochemical methods in conjunction with UV-VIS spectroscopy" needs to be completed with the name/description of the complementary methods. - Fourth paragraph - The sentence "In properly conducted experiments, any mention of chemistry or the presence of nanoparticle residues is deeply unfounded. While potentially valid in some respects, theories aiming to explain UHD effects with some physical or chemical remnants often fall considerably short within this context" is quite controversial. It generates endless discussions since the nanoparticles' rule in UHDs is still unclear. So, it does not help to show the field effects described in the proposed manuscript after a careful, systematic analysis as described. I suggest withdrawing this polemic sentence. - Paragraphs 5 to 8 - Congratulations on coherent domains' clear and didactic definition! - Ninth paragraph - The sentence "(10-2 dilution of the original" needs the ) at the end.   Methods:   - Item 4.2.1. Please give more details about the samples/water storage conditions. They were protected from light, but what about temperature and EMFs? - Item 4.2.2. Please justify the composition of the receiver solution and the role of each component - Item 4.3.2. Please justify the choice of C9 potency and the C12C30C50 potency mix (and why to test this mix) - Item 4.6. The statistical methods are accurate enough.   

Discussion:- Second paragraph: the study design and the interpretation are based on the idea that the signal transmission could be via air or is dependent on the exposition time. However, a third interpretation must be added: depending on the parameter (UV, ORP, and pH), the effect of activation (time 0) could be late, even if mediated via a water/glass medium. The conductivity, instead, seemed to be immediate but transitory. This protocol is not the best for differentiating time from medium when bottles are separated. The time-dependent pH oscillation after the insertion of homeopathic potencies in a polar liquid was previously described in DOI: https://doi.org/10.51910/ijhdr.v21i2.1217. A discussion about these similarities should be added.

Author Response

Please see the submitted review.
